# Comparative analysis of adolescent pregnancy and delivery outcomes versus early adulthood pregnancy in the Asante Akim North District, Ghana

Selina Achiaa Owusu[1]*, Edward Etse Aloryito[2], Esther Sotame[1], Benjamin Asiedu[3]

1 Department of Population, Family and Reproductive Health, School of Public Health, Kwame Nkrumah University of Science and Technology, Kumasi, Ghana, 2 Department of Health Professions Education Innovative Learning, School of Medicine, University of Development Studies, Tamale, Ghana, 3 Department of Business and Social Services Education, University of Cape Coast, Cape Coast, Ghana

* selina.soa@gmail.com

**Data availability statement:** All data set analysed for this study has been uploaded as supporting information.

**Funding:** The authors received no specific funding for this work.

**Competing interests:** The authors have declared that no competing interests exist.

## Abstract

High rate of adolescent pregnancies in Low and Middle-Income Countries (LMICs) has become a major threat to the health and safety of young female adolescents. Despite the evidence of maternal of adolescent pregnancies globally, little is known, especially in low-income countries. This study assessed adolescent pregnancy and delivery outcomes compared to young adults and in the Asante Akim North District, Ghana. Quantitative study was conducted using secondary data on pregnancies and deliveries of teenagers in the Asante Akim North District from 1st January 2018 to 31st December 2020 on 1st November 2021 to 1st December 2021. Health facilities and participants were selected using a multi-stage sampling technique. Data was collected using a structured extraction form and analyzed using descriptive and inferential statistical tools with the help of STATA version 16 at a significant level of 0.05. A total of 3036 records (1144 adolescent records and 1892 early adult records) contributed to the study. Most of the adolescent participants (98.5%) were aged 15–19 years, and had attained primary education (48.9%). The proportion of adolescent pregnancies in the study setting is 13.35%. Adolescent pregnancies were associated with Postpartum Haemorrhage (P-value = 0.001), Uterine Tract Infections (P-value < 0.001), Genital Tract Sepsis (P-value < 0.001), Maternal death (P-value < 0.001) and anaemia in pregnancies (P-value = 0.036). The study found the proportion of adolescent pregnancies and deliveries to be high and therefore requires prompt and effective adolescent reproductive health education.

## Introduction

Young people, simply means individuals in the 10–24 years age group. It is characterized as a 'change period' because there is a sharp change and growth in physical and sexual characteristics in both males and females. The development of sexual features is driven by the increase in sex hormones that arise from sexual desires and the engagement in sexual activities among those in this age category [1]. Among the same category of people is an improved fertility rate

of getting pregnant due to improvement in nutrition and health, a decline in age at which adolescent women start to menstruate and initiate their first sexual activity in recent decades [2]. Since the 19th century, the age at menarche has been dropping at a pace of 2–3 months every decade, resulting in overall decreases of roughly three years.

About half of all adolescent girls and boys in the world have had sexual intercourse at some point in their lives, with the majority initiating sexual intercourse before the age of 13 [3]. Adolescent pregnancies in Ghana have risen significantly from 13.8% in 2008 to 14.2% in 2014. Adolescent pregnancies are predicted in an environment where the median age at sexual debut is 18.4 years, and the proportion of 15 to 19 years old reporting sexual experience climbed from 37.3% in 2008 to 42.7 percent in 2014 [4].

Globally, it is estimated that about 18 million girls under 20 years give birth every year. Sixteen millions of these girls are aged 15–19 years, 2 million are below 15 years, and contribute nearly 11% of births worldwide [5,6]. Although these adolescent pregnancy estimates are global figures, more than 90% of the cases occur in the low- and middle-income countries such as East and Southern Africa, West and Central Africa, Caribbean's, Latin America, and South Asia [5,6]. The prevalence of adolescent pregnancy also varies significantly from country to country; region to region and even community of residence (rural or urban) within the same country [6].

In certain traditional communities, adolescent pregnancy and childbirth is socially desired. However, its numbers are very high and regarded as a special problem as it presents certain social and medical complications. Regarding other problems, adolescent pregnancy presents serious health problems to the adolescent mother and child. That is, it is associated with an increase in preterm labour, increased risk of anaemia, urinary tract infections, pre-eclampsia, hypertension, high rate of caesarean sections, preterm birth, low birth weight, and intrauterine growth restriction among women between aged 10–20 years [7,8].

The adverse perinatal and neonatal outcomes associated with young peoples' pregnancy, which are results of a combination of factors, have been studied from the biological, social, nutritional and healthcare perspectives. To the biologist [9] point of view, most adolescents, especially those in early stages who get pregnant, are biologically immature; that is, they have not fully developed tissues, organs and related parts that will sometimes guarantee safe pregnancy and delivery. To the socialist, teenage pregnancy mothers sometimes lack the needed support and affection, stigmatized because they get pregnant out of wedlock [7]. The perinatal and neonatal adverse outcomes are related to the poor nutritional status of young people during and weeks after pregnancy, according to some group of nutritionists [3], whilst another group believe adverse perinatal and neonatal outcomes are a consequence of inadequate provision of health care, birth prevention and clinical care during pregnancy stage, labour and after labour, directed toward adolescent pregnancy [3].

Adolescents' early pregnancies have major health consequences for teenage mothers and their babies. At least 10 million unintended pregnancies occur each year among adolescent girls aged 10–19 years in the developing world. Complications during pregnancy and childbirth are among the leading cause of death for 15–19-year-old girls globally. Of the estimated 5.6 million abortions that occur each year among adolescent girls aged 10–19 years, 3.9 million are unsafe, contributing to maternal mortality, morbidity and lasting health problems. Adolescent mothers (aged 10–19 years) face higher risks of eclampsia, puerperal endometritis, systemic infections, and anaemia and have a higher rate of cesarean sections than women aged 20 to 24 years. Babies of adolescent mothers face higher risks of low birth weight, preterm delivery and severe neonatal conditions [10–15].

In Ghana, teenagers represent 22% of the total population. The rates of teenage pregnancies are high. Fourteen percent of adolescents aged between 15 and 19 years had begun

childbearing; either they have had a live birth (11 percent) or are pregnant with their first child (3 percent), a slight increase from 13 percent in 2008 [4]. More risks and adverse outcomes accompany pregnancies and deliveries of teenagers compared to older women [16–18]. For this reason, healthcare management in certain parts of the world have special pregnancy care protocols for adolescent pregnancies to curb the morbidity and mortality of adolescent pregnancy complications. However, despite the evidence of the adverse effect of adolescent pregnancies in some studies in other parts of the world [16–20], literature on young people pregnancies in the Asante Akim North Districts is scarce. Therefore, this study seeks to assess pregnancy outcomes and complications among adolescents in the Asante Akim North District, Ghana compared with early adults aged 20–24years.

## Methodology

### Profile of study area and target population

This study was conducted in the Asante Akim North District of Ghana in the Ashanti region. Asante Akim North District was carved out of Asante Akim North Municipal and inaugurated on 28th June 2012. It was created by Legislative Instrument 2057 and has Agogo as its district capital. The district is located in the eastern part of the Ashanti Region and lies between latitudes '60 30' and '70 30' North and longitudes '00 15' and '10 20' West. It shares boundaries with the Sekyere Kumawu District in the north, Kwahu East in the east, Asante Akim South District in the south and the Sekyere East District in the west. It covers a land area of 1,126 square kilometers constituting 4.6% of the Region's land area [4].

The projected population of Asante Akim North District as of 2020 was 83,315 representing 1.4 percent of the Region's total population. About 53.5 percent of dwellings for the population are rural whiles 46.5 percent is urban. Children (42.2%) account for the largest proportion of household members accounting. The Akan culture is the most common in the district. Twi is also the most widely used language of communication in the district. Of the population, 12 years and above, 42.1 percent have mobile phones, and 79.2 percent of the population aged 11 years and over, are literate with 20.8 percent being illiterate [4].

About 69% of the population aged 15 years and older is economically active while 30.8% are economically inactive. Of the population who are economically active, 95.4% are employed while 4.6% are unemployed. Students (47.5%) account for a greater portion of the economically inactive population in the district. 60% of the employed population are engaged as agricultural, forestry and fishery workers while 16.8 percent are service and sales workers [4].

The healthcare system of Asante Akim North District is headed by the Asante Akim North Health Directorate. The Health Directorate is headed by a District Director of Health Services. The healthcare delivery system boasts of 11 health facilities, and two Health Training institutions owned by the Presbyterian Church of Ghana. The health facilities in the district responsible for the healthcare delivery services include Agogo Presbyterian Hospital, Ananekrom Health Centre, Juansa Health Centre, Tanoah Baptist Medical Centre, Amantenaman Health Centre, Akutuase CHPS Compound, Wioso CHPS Compound, Domeabra CHPS Compound, Pekyerekye CHPS Compound, Amantena CHPS Compound and Pataban CHPS Compound [21]. The target population for this study were pregnant young people (10–19years) who constituted the case group of the study and adults aged 20–24 years as the comparison group.

### Inclusion and exclusion criteria

#### Inclusion criteria.

- Only complete data on pregnancies (Information on maternal age, birth weight, Gestational age etc.) of individuals from the ages of 10–19 years and 20–24 years were included.

**Exclusion criteria.**

- Data on pregnancies of individuals older than 24 years were excluded.

- Incomplete data on pregnancies of individuals (Missing information on maternal age, birth weight, Gestational age etc.) from 10–19 years and 20–24 years were excluded.

## Study design and approach

This study adopted a quantitative approach. The cross-sectional study design and approach were chosen to enable the researcher to objectively perform a one-time single-point assessment of the pregnancy outcomes of adolescents in the study setting. The study design again enabled the researcher to make an effective conclusion on the pregnancy outcomes of the case group (adolescents; aged 10–19years) using the comparison group (young adults; aged 20–24 years) as the reference. The cross-sectional study design also strengthened the evidence of association identified between young peoples' pregnancies and pregnancy outcomes.

## Sampling technique and sample size

This study adopted a multi-stage cluster sampling technique to recruit participants for the study. The Asante Akim North District has 13 health facilities made up of 1 district hospital, 1 polyclinic, 3 health centers and 8 CHPS compounds. This study excluded all the CHPS compounds within the district because they do not render pregnancy care and delivery services and, therefore, made it impossible to have data on delivery outcomes. A simple random sampling technique was then used to select 4 of the remaining 5 health facilities for data collection. After selecting the facilities, all records of pregnancies and their related outcomes within the defined period of the study; 1st January 2018 – 31st December 2020 constituted the sample frame of the study. The aggregate of the total number of recorded pregnancies and deliveries, however, constituted the study's sample size.

## Outcomes of interest

**Maternal outcomes of interest.**  Normal delivery, Cesarean section delivery, Anaemia in pregnancy, Pre-eclampsia, Placental abruption, Postpartum haemorrhage, Maternal death, Placental Previa, Gestational Diabetes Mellitus, Pregnancy Induced Hypertension.

**Data collection procedure and tools.**  The researcher developed a data extraction form to record the demographic characteristics of target participants from facility records. The data extraction form was again used to assess the evidence of obstetric, foetal and neonatal outcomes among participants in the study. All facility records on pregnancies from the selected health facilities from 1st January 2018 – 31st December 2020 were retrieved and evaluated. All Antenatal Care (ANC) and Delivery records of the sampled health facilities from 1st January 2018 – 31st December 2020 were retrieved from the archives of the various health facilities. All data on the perinatal outcomes of interest in this study were then extracted and entered into an excel sheet for further data cleaning and analyses. Data collection process started from 1st November 2021 to 1st December 2021.

## Data analyses procedure

This study utilized secondary data (Facility records) for its analyses. Data in this study were analyzed using descriptive and inferential statistical tools. Descriptive tools such as

frequencies with their percentages and tables were used to determine the prevalence of adolescents' pregnancies and the pregnancy outcomes among participants in this study. The prevalence of adolescent pregnancies and deliveries were determined by the percentage of dividing total adolescent pregnancies by the total of female adolescent population in the district. Therefore, the proportion of adolescent pregnancies in the study setting = Total adolescent pregnancies (2018–2020). Total female adolescent population in the district (2018–2020) multiplied by 100%. Inferential statistical tool such as chi-square test was also used to test the association between pregnancy outcomes among the young people using the adult group (20 – 24years) as the reference for comparison. A confidence interval of 95% was used, and a P-value < 0.05 was considered significant in this study. All data analyses in the study were done using STATA version 16.0 software.

## Validity and reliability of the study

This study ensured the study's validity by subjecting all the study methods, procedures and tools for data collection to the critical review and approval of the study supervisor and the ethics committee for content validity. The study methods, procedures and data collection tools were also pre-tested in the Asante Akim South District to correct minor errors before the actual study to ensure the reliability of the study. To further enhance the validity and reliability of the study, the researcher randomly retrieved the mobile numbers of 10% of the study participants and phone calls were made to them. This was to confirm the accuracy of information provided on study participants in their Maternal and Child Health Records Books and what was stated in the records of the health facilities.

## Ethical consideration of the study

Ethical approval for this study was obtained from the Committee of Human Research Publication and Ethics of Kwame Nkrumah University of Science and Technology (REF: CHRPE/AP/457/21) (Appendix B). Site approval for the study was also obtained from the Asante Akim North District Health Directorate before the commencement of the study. The principle of confidentiality and anonymity were strictly adhered to in this study.

# Results

## Socio-demographic characteristics of study participants

The results in Table 1 show that a total of 1,144 adolescent (10–19 years) pregnancies/deliveries and 1,892 pregnancies/deliveries of adults aged 20–24 years, of a total of 8,564 pregnancies/deliveries recorded in the study setting were analyzed. Of the 1,144 adolescent pregnancies/deliveries, the majority occurred among single (674, 58.9%) adolescents aged 15–19 years (1127, 98.5%). Most of the adolescents (559, 48.8%) have had primary school as their highest level of education, followed by JHS/JSS (256, 22.4%), SHS/SSS (222, 19.6), no formal education (80, 7.0%), and post-secondary/vocational education (25, 2.2%). Most of the adolescents had a normal vaginal delivery (1115, 97.5%) with only 29 (2.5%) caesarean section cases. Overall, most of the pregnancies were recorded in 2019 (409, 35.8%) than in 2018 (378, 33.0%) and 2020 (357, 31.2%).

## Prevalence of adolescent pregnancies

According to the District Health Directorate of the Asante Akim North District, the total female adolescent population as at 2020 was 8,564. A total of 1,144 adolescent pregnancies/deliveries were also recorded over the study time frame [21].

**Table 1. Adolescents' maternal characteristics.**

| Variable | Freq (%) |
|---|---|
| **Age range (years)** | |
| 10–14 | 17 (1.5) |
| 15–19 | 1127 (98.5) |
| 20–24 | 1891 |
| **Marital Status** | |
| Single | 674 (58.9) |
| Married | 470 (41.1) |
| Divorced | |
| Widowed | |
| Cohabiting | |
| **Level of Education** | |
| No formal education | 80 (7.0) |
| Primary | 559 (48.9) |
| JHS/JSS | 256 (22.4) |
| SHS/SSS | 222 (19.6) |
| Post-secondary/vocational | 25 (2.2) |
| **Mode of Delivery** | |
| Vaginal | 1,115 (97.5) |
| Caesarean section | 29 (2.5) |
| **Year (Total births)** | |
| 2018 | 378 (33.0) |
| 2019 | 409 (35.8) |
| 2020 | 357 (31.2) |
| **Total** | **1144** |

Therefore, the proportion of adolescent pregnancies in the study setting = Total adolescent pregnancies (2018–2020) = 1,144/ Total female adolescent population in the district (2018–2020) = 9,464 multiplied by 100%.

$$\text{Proportion} = \frac{1144}{8564} \times 100 = 13.35\%$$

Therefore the proportion of adolescent pregnancies in the study setting = 13.35%.

## Maternal outcomes and complications of adolescent pregnancies compared to young adults

The results of the indicators measured in this section were presented under outcome and complication: 7 items on outcome and 6 items on complications. For the entire 3 year period (2018–2020), a substantial number of the adverse cases recorded were among adolescents aged 10–19 years compared with the adult reference group (20–24 years); PPH (18 of 37 cases), mothers admitted to hospital for a longer time (more than 2 days) (41 of 72 cases), genital tract sepsis (24 of 42 cases), Uti/pneumonia, breast infection, infected episiotomy (48 of 79 cases), Maternal death (2 of 3 cases), Anaemia in pregnnacy (188 of 323 cases), and the 'other category' cases (39 of 59 cases).

A chi-square test for association between maternal age and pregnancy outcome among young people and using the adult group (20–24 years) as the reference revealed that most

of the pregnancy outcomes/complications were significantly associated with maternal age. Outcomes including PPH ($X^2$ = 11.573[a], p < 0.001), Admission to hospital for a longer time (more than 2 days) ($X^2$ = 0.264[a], p = 0.033), Genital tract sepsis ($X^2$ = 20.314[a], p < 0.001), UTI/Pneumonia, breast infection, infected episiotomy ($X^2$ = 27.296[a], p < 0.001), Maternal death ($X^2$ = 32.211[a], p < 0.001), Anaemia in pregnnacy ($X^2$ = 4.410[a], p < 0.001), and the 'other outcomes' ($X^2$ = 14.900[a], p < 0.001) were associated with the age of the mother.

However, caesarean delivery: 31 of 72 cases, ($X^2$ = 0.481[a], p = 0.483), and surgical site infection (9 cases of 16), ($X^2$ = 0.137[a], p = 0.277) was not associated with young people pregnancies in this study. Nevertheless, hypertensive disorders of pregnancy (10 of 35 cases), ($X^2$ = 0.152[a], p = 0.505), gestational DM (1 of 10 cases), ($X^2$ = 0.015[a], p = 0.194), premature rupture of the membranes (15 of 29 cases), ($X^2$ = 0.229[a], p = 0.250), and placenta previa: 2 cases, ($X^2$ = 0.030[a], p = 0.184) also had no association with young people pregnancies. These results are presented in Tables 2a, 2b and 2c in S1–S3 Tables (Appendix A).

## Discussion

### Proportion of adolescent pregnancies in the Asante Akim North District

Pregnancy among young women aged 10–19 years continues to be high globally. It has become a global burden due to its influence on adolescent morbidities and mortalities. This study found that adolescent pregnancy/deliveries constituted 13.35% of all pregnancies/deliveries in the district during the 3-year period. This figure may be considered high for a rural district with limited access to healthcare and other social amenities. The high proportion of adolescent pregnancies in the study setting supports claims made by the WHO that, adolescent pregnancies in low- and middle-income countries are high [22,23]. The finding further reiterates the reports of [24] and [2] regarding the high prevalence of adolescent pregnancies in developing countries.

This high prevalence of adolescent pregnancies found in this study may be attributed to poverty and low level of education among young adolescents in the study setting. In a study that found the majority of its study participants having low educational status, it is not surprising that young people pregnancy is high. In a study by [18] it was highlighted that young people living in poverty have lower opportunities for higher education and hence fall victim to unplanned unprotected sex and pregnancies.

The rate at which participants in this study are married raises concerns about early marriages among adolescents in rural communities. In rural communities where access to food, water and shelter are sometimes a major concern, parents of some adolescents may be eager to relieve them from their homes to reduce their burdens. This may contribute to some parents allowing their young girls to go into early dating and early marriages, therefore, contributing to high rate adolescent pregnancies in rural settings [7]. This phenomenon may have contributed to this study's high proportion of adolescent pregnancies.

### Maternal outcomes and complications of adolescent pregnancies in the Asante Akim North District

Globally, complications during pregnancy and childbirth among adolescent women are an established risk factor for many adolescent morbidities and mortalities [25,26]. The study found a significant association between adolescent pregnancies and maternal outcomes and complications such as the development of UTIs (p < 0.001), Genital tract sepsis (p < 0.001), Postpartum Haemorrhage (p < 0.001) and maternal death (p < 0.001) using the outcomes among the comparison group (adults aged 20–24 years) as the reference. Other maternal complications such as anaemia in pregnancy were also associated with adolescent pregnancies

(p = 0.036). In this study, other maternal complications of pregnancies such as delayed first and second stage of labour, post-datism, pre-eclampsia, and breech presentation were also associated with adolescent pregnancies (p = 0.037). The above findings are similar to the findings of [25,26]. It is also worth mentioning that some pregnancy outcomes such as caesarean delivery, hypertensive disorders of pregnancy, gestational DM and placenta previa had no significant association with young people pregnancies.

The adverse maternal outcomes and complications associated with young people pregnancies in this study may be attributed to the fact that most adolescents with poor economic backgrounds and educational status may lack the adequate care needed to ensure their good state of health and hence contribute to these high-risk factors. Therefore, stakeholders must adopt proactive and innovative interventions to prevent and reduce the occurrence of adolescent pregnancies. Adopting adequate sex education, increasing the use of contraception among teenagers at risk of unintended pregnancies and advocating against early marriages are some of the established means to curb young people pregnancies [22]. These may also help to curb the adverse outcomes and complications of adolescent pregnancies in developing countries and communities.

## Weakness of the study

This study is geographically limited to the Asante Akim North District. It is contextually also limited to finding maternal outcomes and complications of adolescent pregnancies. Methodologically, the study could not perform more rigorous inferential analyses to strengthen the evidence of associations between maternal and perinatal outcomes and complications, and adolescent pregnancies. Therefore, it is recommended that further rigorous multivariate regression analyses be done to strengthen the level of evidence between adolescent pregnancies and their associated maternal and perinatal outcomes. A robust methodological design (Cohort) should also be used to enhance the causal relationship between young people pregnancies and perinatal and maternal outcomes.

## Conclusions

From 2018–2020, the total number of young people pregnancies recorded in the Asante Akim North District is 1,144. Most participants are within their late adolescents' ages (15–19 years). The study can conclude that the practice of early marriage in the study setting is high (41.1%). The educational statuses of adolescent participants were low as the majority have either had no formal education or have attained primary or JHS as their highest educational status.

The proportion of adolescent pregnancies in the study setting is about 13.35% of all pregnancies in the district. This rate of adolescents' pregnancies is high and requires stakeholders' prompt and effective intervention.

The study can conclude that maternal health outcomes and complications such as Postpartum Haemorrhage, UTI's, Genital tract Sepsis, maternal death and anaemia in pregnancies are associated with adolescent pregnancies. Other outcomes and complications such as Abruptio placentae, antepartum haemorrhage, delay first and second stage of labour and breech presentation are also influenced by adolescent pregnancies.

## Recommendations to key stakeholders

Considering the results obtained in this study, the recommendations below are made to stakeholders to help contribute to the effective prevention and care for adolescent pregnancies in the Asante Akim North District and Ghana.

- Through the Ministry of Health, the Government of Ghana should collaborate with the Ministry of Education to integrate and intensify adolescent sexual health education in School Health Services in Basic and Senior High Schools. This may further contribute to the provision of Family Planning Services to sexually active adolescents in JHS and SHS sick bays. This will enable adolescents to acquire adequate sexual health knowledge of possible outcomes and complications of adolescents and young adults pregnancies to help them make informed decisions regarding unprotected sex and becoming pregnant.

- Through the various District Health Directorates, Ghana Health Service should intensify community and school-based education on effective contraception among sexually active young people to improve their knowledge and understanding of the best use of contraceptives in order to prevent and reduce the rate of adolescent pregnancies, hence reducing young people pregnancy outcomes and complications.

- Through the various District Health Directorates, Ghana Health Service should provide appropriate training and needed technical support to providers of reproductive health services to reinforce quality reproductive health service delivery. This will help ensure adequate availability of staff and logistics to effectively carry out adolescent counselling and community Information, Education and Communication (IEC) programmes.

- District Health Directorates should partner with NGOs and community leaders to organize frequent community educational programmes to increase knowledge of the perinatal and neonatal outcomes and complications and the socio-economic effect of young peoples' pregnancies. They should again raise awareness on the effects of early marriages, especially in the traditional and rural communities, to reduce the rate of adolescents' pregnancies in the communities. This will further enable adolescents' to also make informed decisions concerning engaging in unprotected sex and therefore becoming pregnant.

- District Health Directorates should collaborate with religious institutions like churches and mosques to educate the youth about abstinence and the risks of unprotected sex and adolescent pregnancies. They should also advise and educate the youth against the effects of adolescents' pregnancies and the need to avoid them.

## Supporting information

**S1 Table.  Maternal outcomes and complications of adolescents and young adults' pregnancies/deliveries.**
(DOCX)

**S2 Table.  Other young people maternal outcomes and/or complications.**
(DOCX)

**S3 Table.  Association between maternal outcomes and complications and maternal age.**
(DOCX)

**S1 Data.  Raw data.**
(XLSX)

## Author contributions

**Conceptualization:** Selina Achiaa Owusu.

**Investigation:** Selina Achiaa Owusu, Esther Sotame.

**Methodology:** Selina Achiaa Owusu, Benjamin Asiedu.

**Project administration:** Selina Achiaa Owusu.

**Resources:** Esther Sotame.

**Supervision:** Esther Sotame, Benjamin Asiedu.

**Validation:** Edward Etse Aloryito.

**Writing – original draft:** Selina Achiaa Owusu.

**Writing – review & editing:** Benjamin Asiedu.

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
