## [Decision Letter · Decision Letter 0]

4 Sep 2024

PGPH-D-24-00278

COMPARATIVE ANALYSIS OF ADOLESCENT PREGNANCY AND DELIVERY OUTCOMES VERSUS EARLY ADULTHOOD PREGNANCY IN THE ASANTE AKIM NORTH DISTRICT, GHANA

Dear Dr. Owusu,

Thank you for submitting your manuscript to PLOS Global Public Health. After careful consideration, we feel that it has merit but does not fully meet PLOS Global Public Health’s publication criteria as it currently stands. Therefore, we invite you to submit a revised version of the manuscript that addresses the points raised during the review process.

We look forward to receiving your revised manuscript.

Kind regards,

Tanmay Bagade, Ph.D., MS (O&G), MPH, MHM

Academic Editor

Journal Requirements:

1. Please provide additional details regarding participant consent. In the ethics statement in the Methods and online submission information, please ensure that you have specified (1) whether consent was informed and (2) what type you obtained (for instance, written or verbal, and if verbal, how it was documented and witnessed). If your study included minors, state whether you obtained consent from parents or guardians. If the need for consent was waived by the ethics committee, please include this information.

Additional Editor Comments (if provided):

Reviewers' comments:

Reviewer's Responses to Questions

**Comments to the Author**

1. Does this manuscript meet PLOS Global Public Health’s publication criteria? Is the manuscript technically sound, and do the data support the conclusions? The manuscript must describe methodologically and ethically rigorous research with conclusions that are appropriately drawn based on the data presented.

Reviewer #1: Yes

Reviewer #2: Yes

2. Has the statistical analysis been performed appropriately and rigorously?

Reviewer #1: Yes

Reviewer #2: Yes

3. Have the authors made all data underlying the findings in their manuscript fully available (please refer to the Data Availability Statement at the start of the manuscript PDF file)?

Reviewer #1: Yes

Reviewer #2: Yes

4. Is the manuscript presented in an intelligible fashion and written in standard English?

Reviewer #1: Yes

Reviewer #2: Yes

5. Review Comments to the Author

Reviewer #1: Thank you for the opportunity to review the paper “Comparative Analysis of Adolescent and Young Adult Pregnancy and Delivery Outcomes in Asante Akim North District, Ghana.” This study examines the prevalence and adverse outcomes of adolescent pregnancies in the Asante Akim North District, Ghana, comparing them to those of young adult pregnancies. It is a cross-sectional study, analyzing secondary data (2018-2020) encompassing 3,036 pregnancy records. The findings reveal that adolescent pregnancies represent 13.35% of all pregnancies in the district, with an association between adolescent pregnancies and adverse outcomes, including postpartum hemorrhage, genital tract sepsis, maternal death, and anemia. The study concludes with a call for enhanced reproductive health education and targeted interventions to reduce the high rates of adolescent pregnancies and their associated complications.

This is an important topic in the maternal health literature. In general, the study is well-conducted and well-presented. However, I have some concerns and recommendations related to methods, as well as a set of minor recommendations related to writing.

Introduction Review

The Introduction provides a comprehensive overview of the context and significance of the study and cites relevant literature to support the study's rationale. It is important to update the literature and better discuss adolescent pregnancy from a gender and life-cycle perspective, considering factors beyond biological aspects. This conceptual approach underscores the notion that age groups are merely operational definitions, downplaying the significant differences in youth experiences across various ages, classes, and social contexts.

Some minor points:

Page 2: Clarify the statement about the decline in age at menarche.

Page 2: Update the statistic on adolescent sexual intercourse or specify the source.

Page 3: Make the transition to global statistics on adolescent pregnancies more seamless.

Methods

The method of this study is generally well-detailed. However, some areas could benefit from additional detail and justification to enhance transparency and address potential biases. The description of the study area and target population is thorough and well-detailed, providing a clear context for the research. To further strengthen this section:

- Including more data on why this particular district was chosen for the study, highlighting any specific issues related to adolescent pregnancies that make this district relevant, will clarify the choice for this specific population.

- Specify the sources of demographic data and ensure they are the most recent and relevant. Justify the age ranges selected for the case and comparison groups. Discuss potential biases introduced by excluding incomplete data and how this was mitigated.

- The inclusion of both maternal and neonatal outcomes provides a holistic view of pregnancy outcomes. However, it is important to define each outcome clearly to avoid ambiguity and discuss any potential measurement biases related to these outcomes.

- The use of a quantitative, cross-sectional study design is appropriate for assessing the association between adolescent pregnancies and their outcomes. The multi-stage cluster sampling technique is a robust method for ensuring a representative sample. Provide more detail on how the random sampling of health facilities was conducted to ensure transparency.

- Clarify the total sample size achieved and its adequacy for statistical analysis.

- Regarding the objective of comparing two distinct age groups, consider using additional statistical methods, such as logistic regression, to control for potential confounders.

Discussion

The discussion section interprets the study’s findings and places them within a broader context, providing insights and practical recommendations. To further strengthen this section:

- Provide additional context and references to support the socio-economic explanations. Include specific data or references regarding the socio-economic conditions in the study area to strengthen the arguments.

- Include a brief discussion on the potential impact of healthcare access and quality on the high prevalence of adolescent pregnancies.

- Enhance the discussion of limitations and potential biases, including the limitations of the cross-sectional design, such as the inability to establish causality.

Reviewer #2: Publication criteria: The manuscript meets PLOS Global Health's criteria as it presents methodologically sound research with appropriate ethical considerations. The conclusions are clearly drawn from the data, addressing an important public health issue in Ghana.

Statistical Analysis: The statistical analysis is rigorous, using appropriate methods to assess the significance of findings, including associations between adolescent pregnancies and adverse outcomes.

Data availability: The authors have made the underlying dat available, supporting transparency and reproducibility of the findings.

Intelligibility and Language: The manuscript is well written, presented clearly, and follows standard English, making it accessible to a broad audience.

Additional comments: The manuscript provides valuable insights into adolescent pregnancy outcomes, but minor revision for clarity in language and expanded explanations in the methodology could further enhance the paper's impact. No concerns regarding dual publication or ethics were noted.

6. PLOS authors have the option to publish the peer review history of their article (what does this mean?). If published, this will include your full peer review and any attached files.

**Do you want your identity to be public for this peer review?** For information about this choice, including consent withdrawal, please see our Privacy Policy.

Reviewer #1: No

Reviewer #2: No

---

## [Editor Report · Decision Letter 1]

11 Oct 2024

PGPH-D-24-00278R1

COMPARATIVE ANALYSIS OF ADOLESCENT PREGNANCY AND DELIVERY OUTCOMES VERSUS EARLY ADULTHOOD PREGNANCY IN THE ASANTE AKIM NORTH DISTRICT, GHANA

Dear Dr. Owusu,

Thank you for submitting your manuscript to PLOS Global Public Health. After careful consideration, we feel that it has merit but does not fully meet PLOS Global Public Health’s publication criteria as it currently stands. Therefore, we invite you to submit a revised version of the manuscript that addresses the points raised during the review process.

EDITOR: Thank you for addressing reviewer's comments.

Please fix a few grammatical errors throughout the manuscript. There are repetitions of words such as "Also". Please rephrase such sentences. 

Recommendations are too lengthy and broad. They should be mentioned before the conclusion section. Please remove the bullet points and summarise your key recommendations and shift them just before the conclusion section. 

We look forward to receiving your revised manuscript.

Kind regards,

Tanmay Bagade, Ph.D., MS (O&G), MPH, MHM

Academic Editor
---

## [Editor Report · Decision Letter 2]

27 Jan 2025

COMPARATIVE ANALYSIS OF ADOLESCENT PREGNANCY AND DELIVERY OUTCOMES VERSUS EARLY ADULTHOOD PREGNANCY IN THE ASANTE AKIM NORTH DISTRICT, GHANA

PGPH-D-24-00278R2

Dear Ms Owusu,

We are pleased to inform you that your manuscript 'COMPARATIVE ANALYSIS OF ADOLESCENT PREGNANCY AND DELIVERY OUTCOMES VERSUS EARLY ADULTHOOD PREGNANCY IN THE ASANTE AKIM NORTH DISTRICT, GHANA' has been provisionally accepted for publication in PLOS Global Public Health.

Best regards,

Julia Robinson

Executive Editor